# Simple and Efficient Confidence Score for Grading Whole Slide Images

**Mélanie Lubrano**[*1,2]                    MELANIE.LUBRANO@MINESPARISTECH.PSL.EU
[1] *Centre for Computational Biology (CBIO), Mines Paris, PSL University, Paris, France*
[2] *Tribun Health, Paris, France*

**Yaëlle Bellahsen-Harrar** [*,3,4]                    YAELLE.BELLAHSEN-HARRAR@APHP.FR
[3] *Service de Pathologie, Hôpital Européen Georges-Pompidou, APHP, France*
[4] *Université Paris Cité, Paris, France*

**Rutger Fick**[2]                                      RFICK@TRIBUN.HEALTH
**Cécile Badoual**[3,4]                              CECILE.BADOUAL@APHP.FR
**Thomas Walter**[1,5,6]                    THOMAS.WALTER@MINESPARIS.PSL.EU
[5] *Institut Curie, PSL University, Paris, France*
[6] *INSERM, U900, Paris, France*

**Editors:** Accepted for publication at MIDL 2023

## Abstract

Grading precancerous lesions on whole slide images is a challenging task: the continuous space of morphological phenotypes makes clear-cut decisions between different grades often difficult, leading to low inter- and intra-rater agreements. More and more Artificial Intelligence (AI) algorithms are developed to help pathologists perform and standardize their diagnosis. However, those models can render their prediction without consideration of the ambiguity of the classes and can fail without notice which prevent their wider acceptance in a clinical context. In this paper, we propose a new score to measure the confidence of AI models in grading tasks. Our confidence score is specifically adapted to ordinal output variables, is versatile and does not require extra training or additional inferences nor particular architecture changes. Comparison to other popular techniques such as Monte Carlo Dropout and deep ensembles shows that our method provides state-of-the art results, while being simpler, more versatile and less computationally intensive. The score is also easily interpretable and consistent with real life hesitations of pathologists. We show that the score is capable of accurately identifying mispredicted slides and that accuracy for high confidence decisions is significantly higher than for low-confidence decisions (gap in AUC of 17.1% on the test set). We believe that the proposed confidence score could be leveraged by pathologists directly in their workflow and assist them on difficult tasks such as grading precancerous lesions.

**Keywords:** Uncertainty estimation, confidence score, grading, multiple instance learning, whole slide images.

## 1. Introduction

Cancer diagnosis usually implies the assessment of biological tissue samples by pathologists. For some cancer types, samples need to be graded according to histological patterns to as-

---

[*] Contributed equally

sess the severity of the disease. This grading task is crucial since it affects the choices in patient's care and follow-up. For head and neck cancers, precancerous lesions (or dysplasia) are graded according to the extent of abnormality detected in the tissue (Gale et al., 2014). However, grading morphological patterns according to their degree of abnormality is a difficult task: WHO nomenclatures try to impose sharp boundaries on a continuous spectrum of lesions, which often leads to poor inter- and intra-grader-reproducibility (Gale et al., 2020; Mehlum et al., 2018).

In the past years, Artificial Intelligence (AI) has been successfully applied to pathology: to detect metastasis (Bejnordi et al., 2017), to determine cancer subtype (Coudray et al., 2018) or to estimate patient prognosis (Courtiol et al., 2019). Several studies even showed the positive benefit that AI tools can have when combined with manual examinations (Kiani et al., 2020; Ba et al., 2022). However, these algorithms are not yet widely accepted for use in clinical practice as standalone diagnostic tools (Van der Laak et al., 2021; Dwivedi et al., 2021) due to concerns about their instability on external cohorts, sensitivity to domain shifts, and lack of interpretability. AI models also deliver their predictions regardless of histologic ambiguity, without providing information about the confidence of the decision, even if the sample is difficult to grade for the pathologist himself. On such ambiguous cases, a pathologist may ask for a second opinion or additional analysis, while AI would not flag the uncertainty.

Measuring Neural Networks (NN) confidence could bring additional information on these difficult cases. Confidence (or equivalently 'uncertainty') of AI models have been explored in the past years (Gal and Ghahramani, 2016; Osband, 2016), and applied to medical context. Still, very little work focused on histopathology applications: its first use was to boost model performances for segmentation tasks (Nair et al., 2020; Camarasa et al., 2020), for active learning (Lubrano di Scandalea et al., 2019) or detection of Out Of Distribution (OOD) samples (Linmans et al., 2020).

Benchmark of the most popular confidence measures were recently conducted: Poceviciute et al. (2022) compared their capacity to identify misclassified patches, and Thagaard et al. (2020) to detect OOD samples at inference. Dolezal et al. (2022) proposed an uncertainty measure robust to domain shift and tested it on cancer subtyping tasks (binary classification of LUAD vs LUSC [1]). However, none of these methods were designed for multiclass problems or grading tasks (ordinal output) and all focused on binary classification of whole slides. In addition, these studies used tile-based methods requiring either local annotation or tile level information (if the disease is uniformly spread on the slide for instance). This paradigm is not suitable for grading tasks because multiple grades of lesions can co-exist on the same slide.

Most of all, AI models for WSI classification are difficult to train due to the large size of pathology datasets. Yet, the most common methods benchmarked in these studies are computationally intensive (relying on generation of a distribution of predictions) and thus not suitable for such datasets containing thousands of slides and millions of tiles.

In this work we propose a measure of uncertainty designed for grading tasks, that does not require extra training or inferences nor particular architecture changes. This uncertainty measure is easily interpretable as it quantifies how much a model hesitates

---

1. LUAD = Lung Adenocarcinoma, LUSC = Lung Squamous Cell Carcinoma

between two classes, and thus particularly suitable for grading tasks. We compared our grade sensitive confidence score with popular ones and showed that our method manages to accurately separate high confidence from low confidence slides and matches the behavior of pathologists. We believe that measuring the confidence of AI models can provide useful information to pathologists, helping them to standardize their diagnoses and increase their trust in AI.

## 2. Related Work

Multiple methods have been proposed to quantify models' confidence. The simplest one consists in considering the softmax output of the model as a direct measure of confidence. This was argued against in (Gal and Ghahramani, 2016) who proposed the Monte Carlo (MC) dropout method as an approximation of a variational Bayes estimator. In MC dropout, multiple forward inferences are generated while keeping the dropout active. The uncertainty is quantified by the variability between the MC samples and measured by the standard deviation of this distribution. Recently, deep ensembles have proven to be better estimates of uncertainty (Dolezal et al., 2022; Poceviciute et al., 2022): multiple networks are trained from different random initializations. Again the uncertainty measure is derived from the distribution of predictions generated. Other techniques have been explored such as Test Time Augmentation (TTA) which consist in applying random transformation to an input image, generating a distribution of predictions. However, the size of histological datasets make such techniques hardly feasible in practice in the field of computational pathology.

## 3. Methods

### 3.1. Weakly-supervised WSI classification

#### 3.1.1. Grading of head and neck precancerous lesions

Grading WSI is a difficult task: often, the borders between each grade are blurry and related histologic patterns overlap. Additionally, several grades of lesions can co-exist on the same WSI. In this study we focused on grading head and neck precancerous and cancerous lesions with respect to their gravity. Such lesions are graded by assessing several morphological and cytological abnormalities of the tissue according to the WHO recommendations (El-Naggar et al., 2017). Head and neck epithelium can be either normal/benign (0), or present a low grade lesion (1), a high grade lesion (2) or have infiltrative carcinoma (3). (See Appendix A.1 for examples).

#### 3.1.2. Dataset

To assess the benefit of our confidence measure in the context of a grading task we used an in-house dataset of head and neck precancerous and cancerous tissue samples obtained from the HEGP pathology department. The dataset was composed of 2121 Hematoxylin and Eosin (H&E) slides. Each slide was labeled with the most severe lesion it contained (grade 0 to 3). Considering the inherent ambiguity of these histologic patterns, a gold standard test set of 128 slides was blindly reviewed by 2 pathologists and consensus diagnosis was found. An extensive description of the dataset can be found in (Lubrano et al., 2022).

### 3.1.3. Attention MIL Architecture

We used an architecture derived from the Attention MIL model proposed by (Ilse et al., 2018). Each WSI is cut into small tiles and fed to a NN. A global label associated with the WSI is used to train the model. An attention mechanism allows to detect the most relevant tiles for the classification. Details of the architecture can be found in Appendix A.2.

### 3.1.4. Cost sensitive training

To leverage the ordinal nature of the class we used the cost aware classification loss introduced in (Chung et al., 2015): the Smooth One Sided Regression (SOSR) loss. The output of the network is a class specific risk $\hat{C}$ rather than a posterior probability. No softmax is used. The predicted class corresponds to the class minimizing this risk. The loss is defined as follows:

$$\mathcal{L}_{SOSR} = \sum_i \sum_j ln(1 + exp(\mathbf{2}_{i,j} \cdot (\hat{c}_i - \mathcal{C}_{i,j}))) \tag{1}$$

With $\mathbf{2}_{i,j} = -\mathbf{1}_{i \neq j} + \mathbf{1}_{i=j}$ , $\hat{c}_i$ the $i$-th coordinate of the network output and $\mathcal{C}$ the error table. The use of ordinal risk predictions is a crucial requirement for pathologists as it penalizes errors with high medical impact. In contrast, if we used a normal classification setting with cross-entropy, severe misclassifications would be likely to be more frequent. In this paper, we defined the error table as the one proposed by the TissueNet Challenge (Loménie et al., 2022). It can be found in the Appendix A.3.

## 3.2. Confidence measures

### 3.2.1. Our method: grade sensitive confidence score

Our confidence score was derived from the risk estimation vector outputted by the last layer of the network. The softmax of the inverted risk (- risk vector) was computed, turning cost estimation into probabilities. This makes the score easier to interpret and to compare, while not changing the ordering of values. The confidence score was defined as the difference between the two highest risk probabilities: if the probabilities were close, the network was hesitating between two classes, if they were far, the network was considered more confident (see Algorithm 1).

---

**Algorithm 1:** Grade sensitive confidence score (ours)

---
**Input:** $Y \in \mathbb{R}^n$, with $n$ the number of classes
**Output:** $u$, the confidence score
$Y \leftarrow softmax(-Y)$;
$Y \leftarrow sort(Y, ascending = False)$;
$\mathcal{U} \leftarrow Y_0 - Y_1$;

---

We benchmarked our method against the most popular ones in medical image analysis (MC Dropout and Deep Ensembles). As a baseline we considered the raw output vector of the NN as a confidence measure.

### 3.2.2. MC Dropout

Our model was trained with dropout layers with rate r. At inference, $M$ forward passes with dropout layers enabled were performed as described in (Gal and Ghahramani, 2016). The confidence was derived from the standard deviation of the generated Monte-Carlo samples according to the equation 2. The standard deviation was inverted and normalized between 0 and 1 to be compared with the grade sensitive confidence measure.

$$\mathcal{U}_{unormalized} = \sqrt{\frac{\sum_{m=0}^{M}(Y_m - \bar{Y})}{M - 1}} \qquad (2)$$

With $Y \in \mathbb{R}^n$ the output vector, $n$ the number of classes, $M$ the number of MC samples.

### 3.2.3. Deep ensembles

$D$ networks were initialized with different seeds and trained independently from each other. The confidence score was computed from the standard deviation of the $D$ predictions made from the different networks similarly as Equation (2). Again, for comparison purposes, the standard deviation is inverted and normalized between 0 and 1.

### 3.3. Implementation and training details

The dataset was split in 5 training and validation sets in a stratified way to perform a 5-folds cross validation. The folds were used for training and hyper-parameter optimization. The gold standard test was only used for evaluation purposes. WSI were cut in tiles of 224x224 pixels without overlap at resolution of 10X ($1\mu m$/pixel) and background tiles were removed, resulting in a total of 3.9 million of tiles (from 2121 slides). DenseNet121 was used to extract features from the tiles. This convolutional part of the network was frozen and initialized with pre-trained weights obtained from self-supervised learning. The self-supervised architecture relied on SimCLR method (Chen et al., 2020) and is described in Appendix A.4. Dropout rates were set to 0.5. Deep Ensembles were obtained by training 5 models ($D = 5$) with different seeds for the 5 data folds. 50 MC samples were generated ($M = 50$) for the MC dropout method. Predictions were made on the gold standard test set by averaging the output vectors of the $D$ inferences (for deep ensembles) or the $M$ inferences (for MC dropout) and then averaged on the 5 folds. Models were trained for 100 epochs with early stopping on the validation sets. RMSProp optimizer was used with a momentum of 0.5 and a learning rate of 1e-4. At each training step, all the tiles of the randomly sampled slide were used. The implementation was done with Tensorflow (TF 2.8).

## 4. Experiments and results

### 4.1. Capacity to detect difficult slides

To compare the confidence measures capacity to detect the most ambiguous slides, we evaluated the classification performances on the gold standard test set for a range of confidence thresholds. As the threshold increased, more slides were considered uncertain by the model and removed from the test set. Metrics were then computed on the remaining slides. The

baseline consisted in the raw risk estimates obtained when training with the SOSR loss. It was compared with the deep ensembles, the MC dropout, and our proposed grade sensitive measure. Comparison was also made with the evolution of the AUC when slides were picked randomly (Figure 1). We observed that the grade sensitive measure, raw risk and deep ensembles have similar behaviors: they consistently identify more difficult slides on which the model has more chance to make a misprediction. On the contrary, MC dropout fails to identify the most uncertain samples: the AUC is flat when removing the first slides (the most uncertain according to the MC Dropout measure), similar to random removal of slides.

.

Figure 1: Overall AUC evolution on the Gold Standard Test set as we remove the most uncertain slides. We measured the overall AUC (average on the 4 classes) on the same number of slides, after removing the most uncertain slides of the test set. The faster the AUC increases, the better the selected slides are. Per class performances can be visualized in Appendix B.4. Evolution with respect to the confidence score threshold is available in Appendix B.2

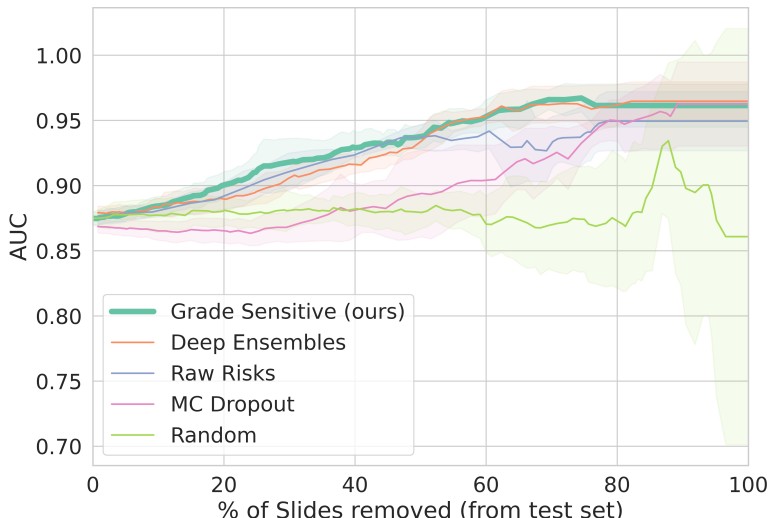

## 4.2. Classification performances at a particular confidence level - Screening use case

In the setting of a screening use-case (e.g. to use AI help only if the prediction is reliable), a confidence threshold has to be set. In real life cases the threshold would be set on the validation sets, based on clinical constraints: to reach a Negative Predictive Value (NPV) on the abnormal (dysplasia + carcinoma) class > 0.95 for instance. Here, to compare the metrics on the same amount of slides and ensure a fair comparison between sets and methods, we defined the threshold as the median of the confidence level. The distribution

of confidence scores across classes revealed that confidence levels were always high for the classes 0 and 3, that are indeed the most simple classes to detect for pathologists. On the other hand, confidence levels are often lower on intermediary classes (1 and 2) which are more difficult to grade and the criteria for each class are difficult to assess (Appendix B.3). In Table 1, we observe that the grade sensitive measure leads to the larger gaps between the low and high confidence subsets of slides.

Table 1: AUC for the stratified gold standard test set in high confidence and low confidence slides. Predictions on the test set are obtained by averaging the predicted probabilities from the 5 trainings on the different folds (ensembling) and bootstraping was conducted to compute final metrics. Other metrics and confidence intervals are available in Appendix B.5

| threshold= median | AUC (Average on 4 classes) | | | NPV Carcinoma (3) | | | NPV Abnormal (1+2+3) | | |
|---|---|---|---|---|---|---|---|---|---|
| *1000 bootstraps* | *Low Confidence* | *High Confidence* | *Gap* | *Low Confidence* | *High Confidence* | *Gap* | *Low Confidence* | *High Confidence* | *Gap* |
| **MC-Dropout** | 0.842 | 0.884 | 4,2% | 0.950 | 0.936 | -1,4% | 0.664 | 0.733 | 6,90% |
| **Raw risk (SOSR)** | 0,797 | 0,934 | 13,70% | 0,920 | **1.0** | **8,0%** | 0,621 | 0,742 | 12,10% |
| **Deep Ensembling** | 0.790 | 0.928 | 13,8% | 0.938 | **1.0** | 6,2% | 0.540 | 0.809 | 26,90% |
| **Grade Sensitive (Ours)** | 0.770 | **0.941** | **17,1%** | 0.920 | **1.0** | **8,0%** | 0.496 | **0.867** | **37,10%** |

### 4.3. Correlation with histologic ambiguity

We compared the level of confidence with the results of the dual blind review of the gold standard test set. Confidences were measured for slides on which pathologist agreed in first place and slides on which pathologist disagreed during the blind review. Confidence scores for deep ensembles and grade sensitive methods were significantly correlated with the review: higher confidence were associated with slides easier for pathologists (the one they agreed on) and lower confidence on more difficult slides (the one they disagreed on). Figure 2 suggests that the confidence level of the model correlates with the risk of disagreement between reviewers. We note that the high variability of confidence scores are expected. Indeed, if the grading of a slide was so difficult that the assignment was random, there would still be a 25% chance that the reviewers agree. If in this case the slide was given a low confidence by the score, it would lead to a low confidence score for a slide the reviewers agreed on. As the confidence scores are measuring different aspects of the prediction process (distance measure, variance measure, risk measure), they are expected to behave differently and vary on different ranges, but this has no negative impact on the quality of the score.

## 5. Discussion and Conclusion

We benchmarked our grade sensitive confidence score with popular uncertainty measures for the difficult task of grading precancerous lesions on histopathological samples. We showed that the grade sensitive score, as well as deep ensembles scores allowed us to accurately detect poor confidence samples while MC dropout measure was actually not performing well in the grading setting (Fig. 1). This observation corresponds to what is found in the literature (Thagaard et al., 2020; Poceviciute et al., 2022). Additionally, our method

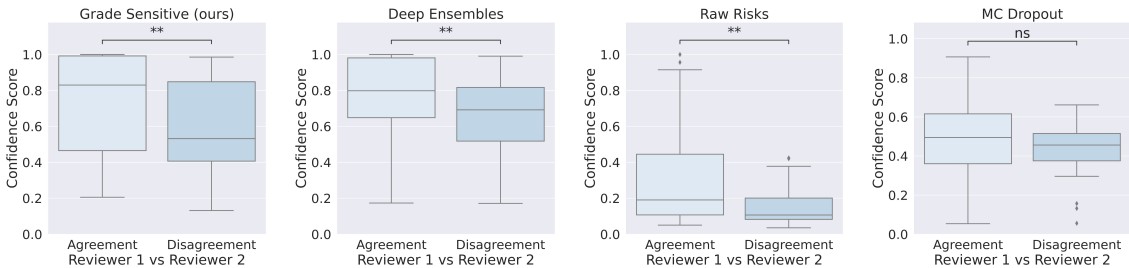

Figure 2: Confidence score distributions compared to pathologist agreements

led to larger gains in classification performance between low and high confidence scores (Table 1), demonstrating that it is the most appropriate method for clinical practice among the techniques tested.

Our grade sensitive method is computationally efficient and is suitable to be used in weakly supervised workflows as it does not requires any annotations. In contrast to deep ensembles that require the training of an extra 4 models to assess the confidence, multiplying by such the training and prediction time. Given that WSI classification typically relies on large datasets and requires long training times, this is an important advantage of our method. Surprisingly, the raw risk estimates were a good assessment of the confidence, managing to stratify the test set between high and low confidence subsets almost as well as deep ensembles. This goes against accepted theory that softmax output of a network is not a good uncertainty estimate (Gal and Ghahramani, 2016; Pearce et al., 2021). We believe this contradictory behavior can be caused by the grading setting. Indeed, in other studies, uncertainty estimates were always computed for binary tasks. In such setting, the softmax layer inevitably tends to force extreme decisions, the subtleties needed to stratify the data according to their confidence are erased by this excessive behavior. On the other hand, in the context of grading, softmax can have more difficulties forcing a class to a probability of 1, leading to a better distribution of probabilities that can thus be used more efficiently to measure a models confidence.

We identified that the most uncertain slides according to our measure were mostly low grade (1) and high grade lesions (2), ie. intermediate classes. This observation corresponds to what pathologists experiment in practice. Indeed, the grading system of head and neck precancerous lesions has always been controversial (Gale et al., 2020; Mehlum et al., 2018), and no less than 4 grading nomenclature have been proposed in the last decades. This illustrates the inherent difficulty of the grading task, where borders between classes are subject to noise and pathologists' own calibration.

Finally, we showed that grade sensitive, raw risks and deep ensembles measures were significantly correlated with pathologists slides agreements (Figure 2). This interesting finding demonstrates the portability of such measures to clinical practice. Indeed, the confidence is able to detect slides on which pathologists will also struggle, and probably disagree. This can be of great help for detecting more difficult slides and either filter them out or directly ask for a second opinion.

In conclusion, we proposed new approach for evaluating the reliability of AI models for WSI grading. This approach is easy to understand, cost-effective, and takes into account the inherent uncertainty of histological samples. We believe this contribution is an interesting step toward more reliable AI model to be used in clinical practice, especially for subjective tasks such as grading precancerous lesions.

## Acknowledgments

ML was supported by a CIFRE PhD fellowship founded by TRIBUN HEALTH and ANRT. This work was supported by the French government under management of ANR as part of the "Investissements d'avenir" program, (ANR-19-P3IA-0001,PRAIRIE 3IA Institute).

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

## Appendix A. Methods

### A.1. Grading of head and neck precancerous lesions

WSI - with global label:
Carcinoma (3)

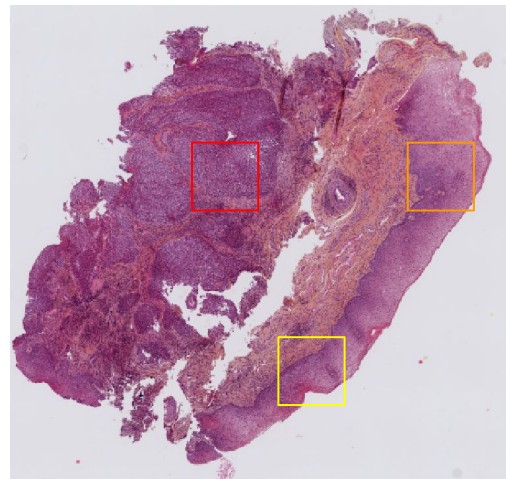

Example of patterns:

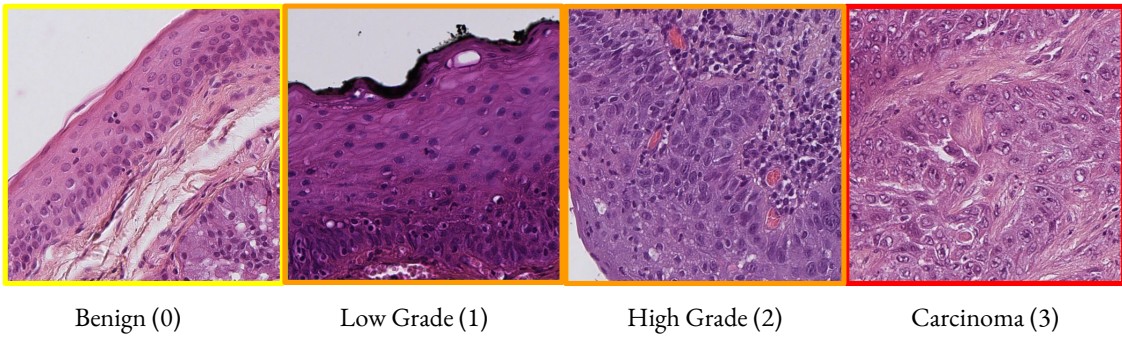

| Benign (0) | Low Grade (1) | High Grade (2) | Carcinoma (3) |

Figure 3: Examples of WSI and lesions according to the grade (20X): the lesions affects the surface epithelium. The lesion grade is determnined taking into account (but not exclusively) the thickness of epithelium affected by dysplastic patterns.

### A.2. Architecture Details

Table 2: Whole Slide Classification - Summary Table

| Layers | Type |
|---|---|
| **Feature Extractor** | DenseNet121 [1024] |
| **Attention-MIL** | **Dimensionality Reduction** |
| | FC [128] + tanh |
| | Dropout |
| | **Tile Scoring** |
| | FC [128] + softmax |
| | Dropout |
| | **Classification** |
| | FC [200] + relu |
| | Dropout |
| | FC [100] + relu |
| | Dropout |
| | FC [4] |

## A.3. Cost Matrix

Misclassification errors do not lead to equally serious consequences. Accordingly, a panel of pathologists established a grading of each of these errors i.e they attributed to each pair of possible outcome $(i, j) \in \{0, 1, 2, 3\}^2$ a severity score $0 \leqslant C_{i,j} \leqslant 1$ (Table 3).

Table 3: Weighted Accuracy Error Table - Error table to ponderate misclassification according to their gap with the ground truth

| Ground Truth | Benign (pred) | Low-grade (pred) | High-grade (pred) | Carcinoma (pred) |
|---|---|---|---|---|
| Benign | 0.0 | 0.1 | 0.7 | 1.0 |
| Low-grade | 0.1 | 0.0 | 0.3 | 0.7 |
| High-grade | 0.7 | 0.3 | 0.0 | 0.3 |
| Carcinoma | 1.0 | 0.7 | 0.3 | 0.0 |

## A.4. Self-supervised Implementation Details

Our Self-supervised model relied on SimCLR architecture. It consisted of a feature extractor (DenseNet121) and a projection head (3 dense layers). The model was trained on 3.5 million tiles of 336x336 pixels from the dataset, using a batch size of 864, a temperature of 0.1, and a learning rate of 10e-4. The feature extractor was initialized with ImageNet pretrained weights and trained for 5 epochs with the feature extractor frozen. After this, the model was trained for a total of 300 hours (143 epochs) using augmentations such as cropping and resizing the tiles to 224x224 pixels, random rotations of 90 degrees, flipping, custom stain augmentation, and color jittering to adjust brightness, hue, contrast, and saturation.

# Appendix B. Additional Results

## B.1. Capacity to detect difficult slides

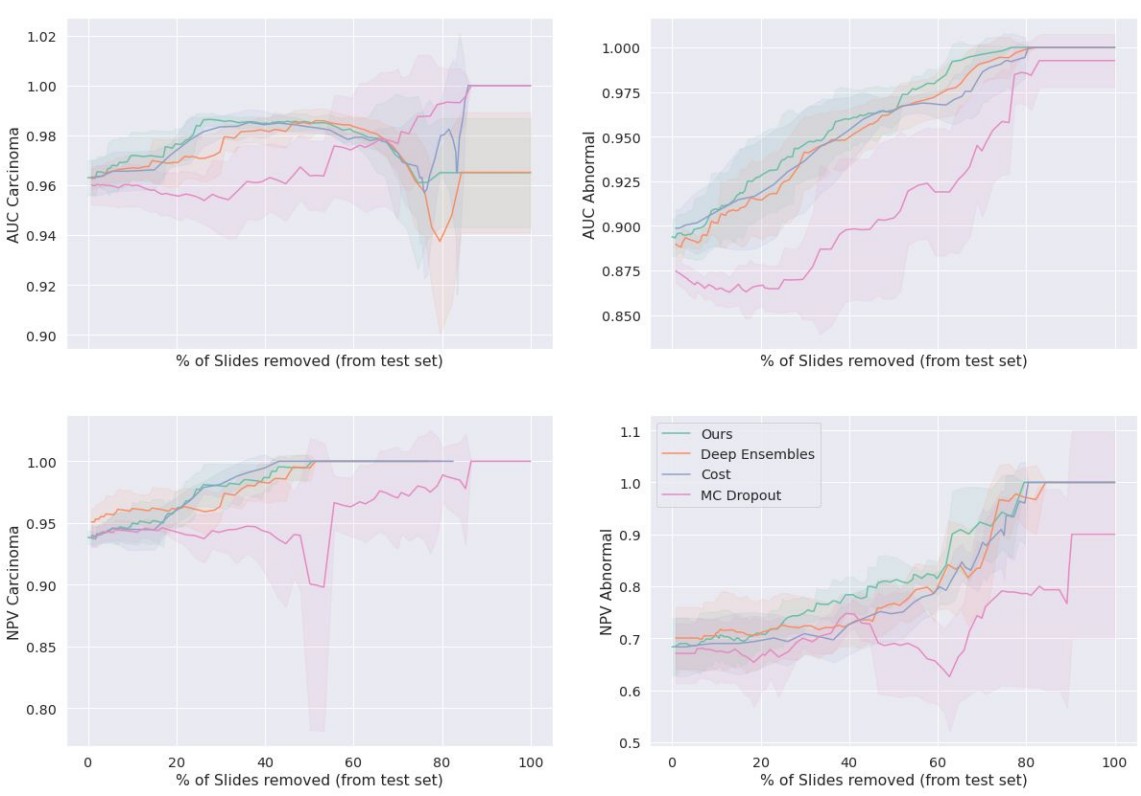

Figure 4: Evolution of various metrics on the Gold Standard Test set as we remove the most uncertain slides

### B.2. Overall AUC evolution with respect the confidence threshold

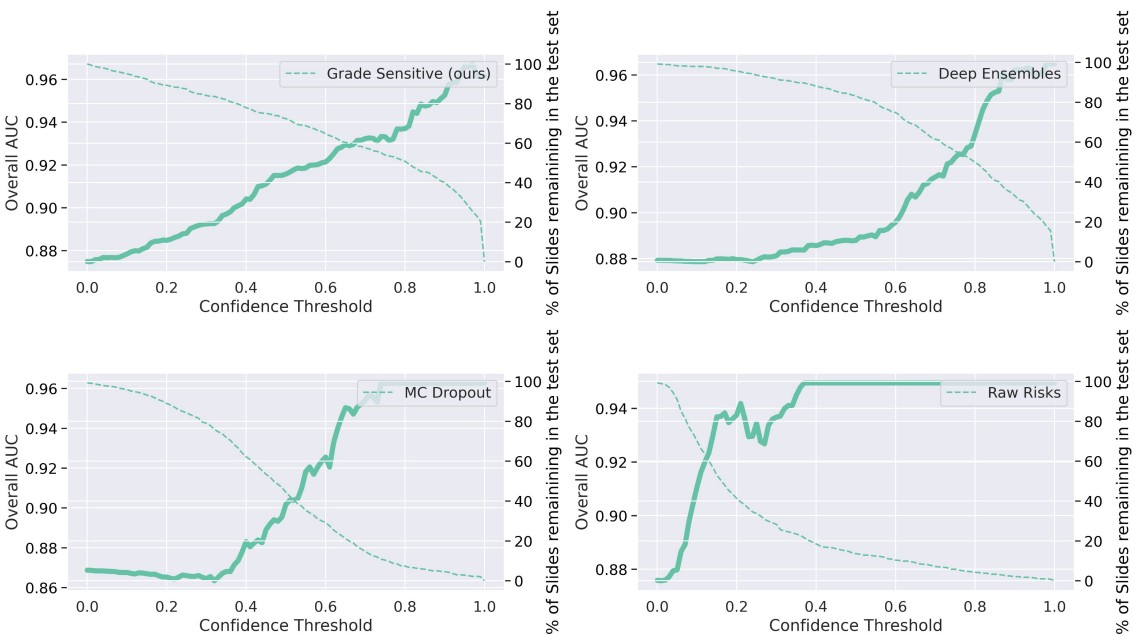

Figure 5: Evolution of the Overall AUC and the number of slides removed with respect to the confidence score threshold. Dashed line are link to the right axis (% slides remaining in the test set), et full lines are linked to the left axis (Overall AUC). This representation allows us to verify that not all the slides are removed immediately when increasing the threshold.

## B.3. Confusion Matrices and Confidence score distributions

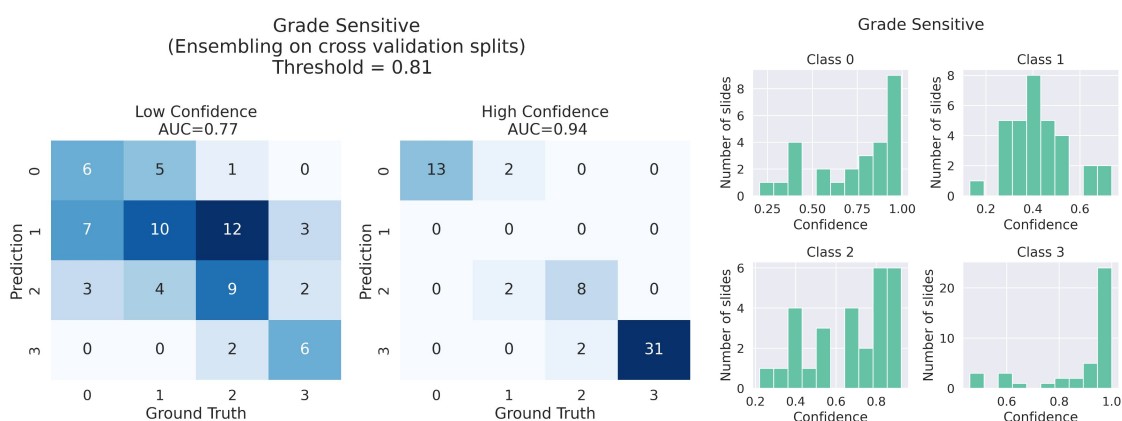

Figure 6: Confusion Matrices on high and low confidence (threshold=median) based on grade sensitive measure (left) and distribution of confidence scores for each class (right)

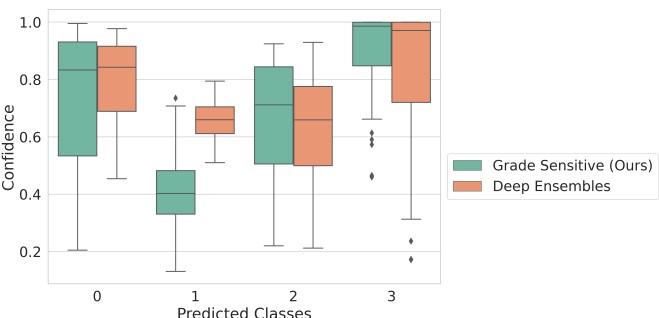

Figure 7: Confidence level per class for grade sensitive and deep ensembles methods. The confidence is lower for intermediary classes 1 and 2.

Table 4: Average Confidence values per slides for each method

| Classes | Grade Sensitive (ours) | Deep Ensembles | MC Dropout | Raw Risk |
|---|---|---|---|---|
| **Benign (0)** | 0.74 | 0.79 | 0.47 | 0.25 |
| **Low Grade (1)** | 0.42 | 0.66 | 0.45 | 0.08 |
| **High Grade (2)** | 0.66 | 0.63 | 0.47 | 0.13 |
| **Carcinoma (3)** | 0.89 | 0.82 | 0.50 | 0.46 |

## B.4. ROC Curves

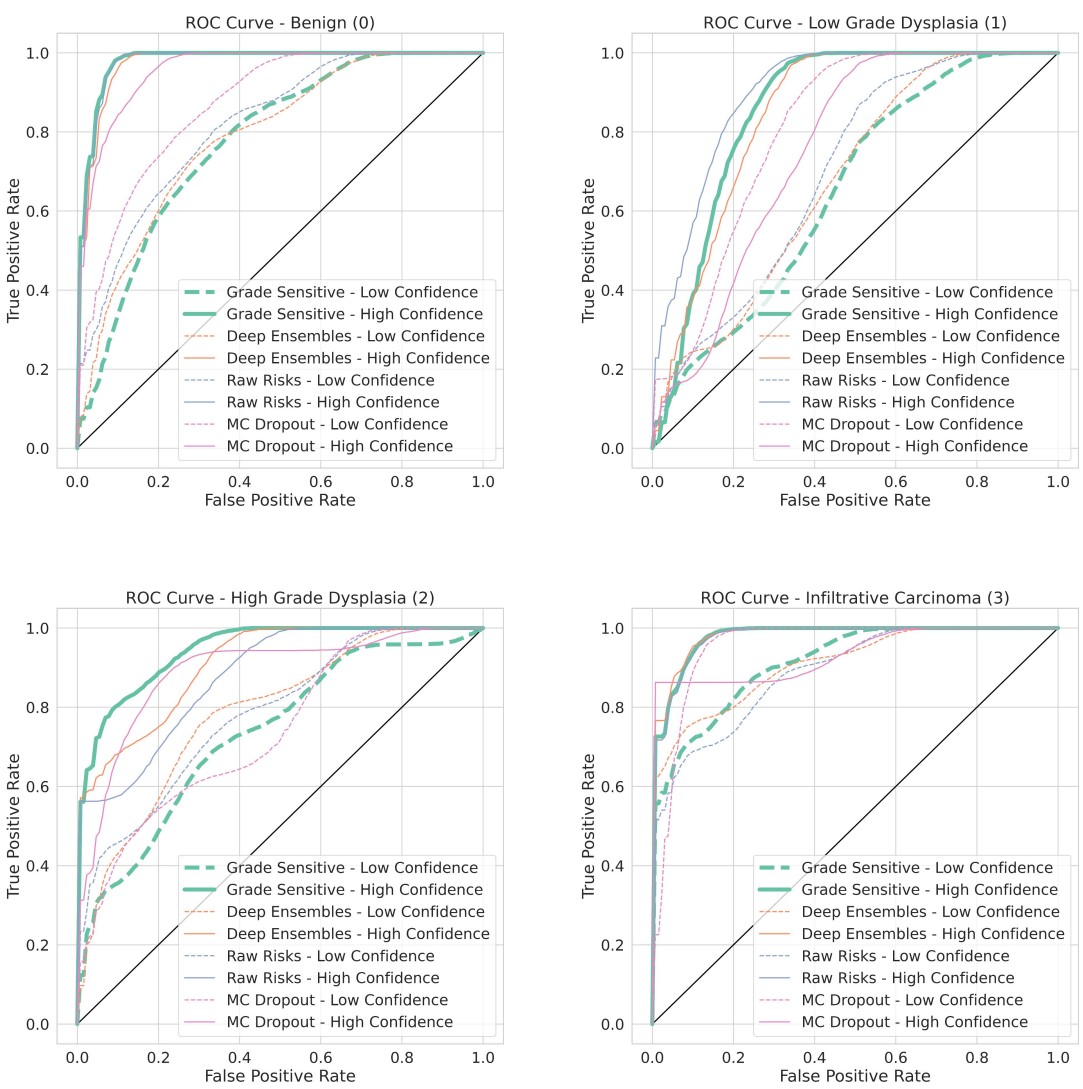

Figure 8: ROC AUC on the Gold Standard Test set after dichotomized between High Confidence slides and low confidence slides (threshold = median, 10 000 bootstraps)

### B.5. Stratification of the test set between high and low confidence slides

Table 5: Stratification of the gold standard test set between high confidence and low confidence slides. Ensembling predictions from the 5 folds cross validation trainings and bootstraped.

| threshold= median | AUC (Average on 4 classes) | | | NPV Carcinoma (3) | | | NPV Abnormal (1)+(2)+(3) | | |
|---|---|---|---|---|---|---|---|---|---|
| *1000 bootstraps* | *Low Confidence* | *High Confidence* | *Gap* | *Low Confidence* | *High Confidence* | *Gap* | *Low Confidence* | *High Confidence* | *Gap* |
| **MC-Dropout** | 0.842 [0.78, 0.9] | 0.884 [0.823, 0.938] | 4,2% | 0.95 [0.87, 1.0] | 0.936 [0.86, 1.0] | -1,4% | 0.664 [0.364, 0.917] | 0.733 [0.47, 1.0] | 6,9% |
| **Raw risk (SOSR)** | 0,797 [0.719, 0.862] | 0,934 [0.882, 0.978] | 13,70% | 0,92 [0.847, 0.984] | **1 [1.0, 1.0]** | **8,0%** | 0,621 [0.25, 1.0] | 0,742 [0.524, 0.938] | 12,1% |
| **Deep Ensembling** | 0.79 [0.719, 0.858] | 0.928 [0.879, 0.978] | 13,8% | 0.938 [0.879, 0.985] | **1.0 [1.0, 1.0]** | 6,2% | 0.54 [0.222, 0.819] | 0.809 [0.588, 1.0] | 26,9% |
| **Grade Sensitive (Ours)** | 0.77 [0.689, 0.84] | **0.941 [0.902, 0.977]** | **17,1%** | 0.92 [0.848, 0.983] | **1.0 [1.0, 1.0]** | **8,0%** | 0.496 [0.214, 0.778] | **0.867 [0.667, 1.0]** | **37,1%** |

Table 6: Other metrics

| threshold= median | AUC Carcinoma (3) | | | AUC Abnormal (1)+(2)+(3) | | |
|---|---|---|---|---|---|---|
| *1000 bootstraps* | *Low Confidence* | *High Confidence* | *Gap* | *Low Confidence* | *High Confidence* | *Gap* |
| **MC-Dropout** | 0.953 [0.893, 0.997] | 0.936 [0.857, 1.0] | -1,70% | 0.825 [0.708, 0.919] | 0.948 [0.889, 0.99] | 12,30% |
| **Raw risk (SOSR)** | 0,891 [0.768, 0.988] | 0,979 [0.939, 1.0] | **8,80%** | 0.737 [0.577, 0.875] | 0,968 [0.922, 1.0] | 23,10% |
| **Deep Ensembling** | 0.911 [0.8, 0.991] | **0.982 [0.942, 1.0]** | 7,10% | 0.722 [0.567, 0.849] | 0.968 [0.916, 1.0] | 24,60% |
| **Grade Sensitive (Ours)** | 0.915 [0.815, 0.989] | 0.98 [0.942, 1.0] | 6,50% | 0.718 [0.562, 0.852] | **0.974 [0.931, 1.0]** | **25,60%** |

### B.6. Analysis on most hesitating slides, comparison between ordinal loss and cross-entropy

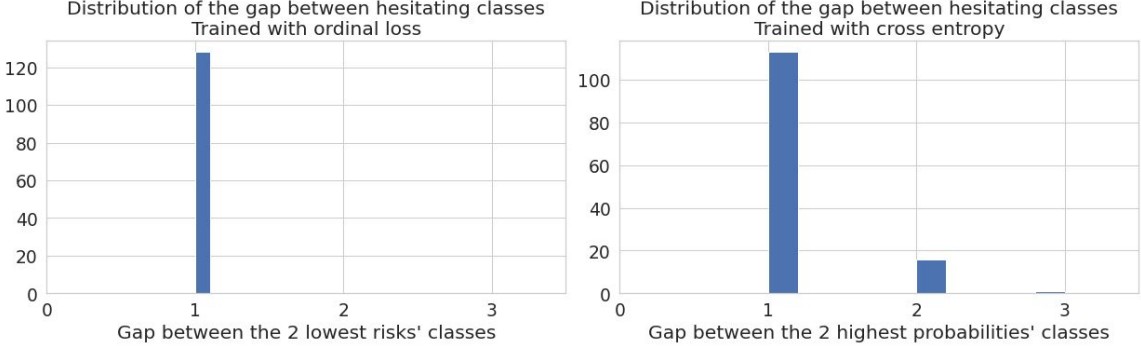

Figure 9: When trained with cross entropy (right), the model hesitates between classes that are not always adjacent. On the other hand, when trained with the ordinal loss (left), the gap between the most probables classes (or less risky) is always of 1. This behavior align with pathologists experience and biological groundings. Predictions on the gold standard test set were obtained by ensembling on the 5 folds.

### B.7. Comparison with different cost matrices

The cost matrix defined in the TissueNet Challenge (Loménie et al., 2022) was set by an expert consensus within a scientific council. It reflects the medical impact a diagnosis error

can have on the patient's care, cure and follow up. For instance, according to experts, diagnosing a benign case (0) instead of a low grade lesion (1) has less impact than diagnosing a high grade (2). Without any prior knowledge or experts to set error costs, one could use more standard matrices with linear or quadratic weights for instance. The following results compare performances obtained from training with the custom costs matrix and a linear cost matrix as defined below.

$$\text{Linear cost matrix: } \begin{bmatrix} 0.0 & 0.33 & 0.66 & 1.0 \\ 0.33 & 0.0 & 0.33 & 0.66 \\ 0.66 & 0.33 & 0.0 & 0.33 \\ 1.0 & 0.66 & 0.33 & 0.0 \end{bmatrix}$$

Table 7: Per class classification performances (avg +/- std on the 5 folds) on the gold standard test set

|  | Normal (0) | Low Grade (1) | High Grade (2) | Carcinoma (3) |
|---|---|---|---|---|
| **AUC Custom Matrix** **Overall AUC= 0.875 +/- 0.005** | 0.911 +/- 0.0078 | 0.78 +/- 0.0101 | 0.845 +/- 0.007 | 0.963 +/- 0.0071 |
| **AUC Linear Matrix** **Overall AUC= 0.879 +/- 0.0064** | 0.91 +/- 0.0047 | 0.795 +/- 0.0171 | 0.849 +/- 0.0099 | 0.963 +/- 0.0068 |

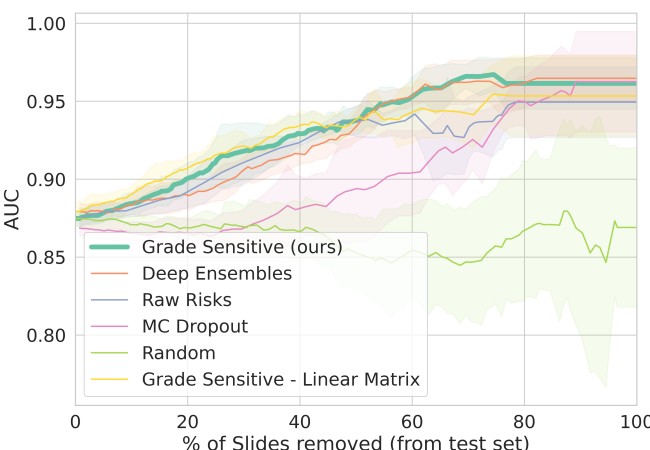

Figure 10: Similar to Figure 1. Overall AUC evolution on the Gold Standard Test set as we remove the most uncertain slides. Comparison with trainings done using a linear cost matrix in yellow.

