# OpenReview forum: "Simple and Efficient Confidence Score for Grading Whole Slide Images"
_MIDL.io/2023/Conference — MIDL 2023 Oral_

### Official Review · Reviewer_AXSV · 2023-02-01

**Confidence:** 4
**Preliminary Rating:** 3
**Recommendation:** Poster

**Summary:**

This papers proposes a method for uncertainty estimation for deep learning models. The proposed method is particularly suited for the task of grading of histopathology image analysis as 1) it works in combination with ordinal regression loss, that is suitable for grading tasks and 2) does not require multiple runs of inference, which can be expensive for large histopathology slides.

**Strengths:**

The main strength of the paper are:
- The relatively straightforward approach that can be easily implemented, independent of the model architecture.
- Good evaluation approach that both uses the uncertainty estimation to reject part of the dataset with high uncertainty and correlates the uncertainty with grading uncertainty during the annotation process.

**Weaknesses:**

The presented approach for uncertainty estimation can in principle also work with cross-entropy loss. I find it to be an omission that the authors did not present results with cross entropy loss as well. The ordinal regression loss is less commonly used, and is perhaps more difficult to optimise than cross-entropy (I do not claim this to be the case, but it should be at least commented).

**Deanonymize Review:**

no

**Detailed Comments:**

It would be also interesting to see experiments with more cost matrices (e.g. linear and quadratic cost). I find the cost defined by pathologists a bit arbitrary.

**Paper Type:**

methodological development

**Questions To Address In The Rebuttal:**

Experiments with cross-entropy loss will significantly improve the quality of the manuscript. At this point, it is uncertainty if the ordinal loss is more difficult to optimise than cross-entropy. Experiments with different cost matrices would be a plus.

---

### Official Review · Reviewer_1zEe · 2023-02-01

**Confidence:** 4
**Preliminary Rating:** 4
**Recommendation:** Poster

**Summary:**

The authors propose a method to compute uncertainty scores for a grading task with ordinal outputs, specifically grading precancerous lesions on whole slide images. The method relies on estimating the risk at the final layer of the Deep Network instead of softmax probabilities and minimizing the smooth one sided regression loss. The confidence is calculated as the difference between the top two softmax probabilities of the negated risks among all grades. This method outperforms Monte Carlo Dropout method and performs on par with Deep Ensemble uncertainty estimation method. Further, the score reflects the pathologists’ uncertainty on a data point, as validated by an experiment conducted on data labelled by two pathologists.

**Strengths:**

1. The proposed method performs better or on par with the prior methods while also being computationally more efficient. The methods are evaluated on data reviewed by two pathologists.
2. The authors have addressed the relevant prior and compared their method with the prior work.
3. The methods and results are clearly explained.

**Weaknesses:**

1. The proposed confidence score is difference of probabilities and not normalized, whereas the MC dropout and Deep Ensemble confidence scores are normalized between 0 and 1.
2. In Fig 1., authors show the evolution of accuracy as a function of % of slides removed. It is not shown how the % of slides removed or accuracy varies with the threshold for selecting uncertain slides.
3. In Fig. 2, the authors justify the large variance shown by the proposed method. However, this does not adequately explain why the other methods including raw risks show much lower variance compared to the proposed method.
4. The authors claim that the most uncertain slides correspond to grade 1 and 2 lesions. It would be helpful to have some numbers to support this claim. For example, the average confidence scores across samples of each grade.
5. Do the MC dropout and deep ensemble networks output probabilities or risks? In case they compute risks, why is softmax not applied to the inverted risks (similar to the proposed method)?

**Deanonymize Review:**

no

**Detailed Comments:**

Citation missing for uncertainty estimation with Test Time Augmentation in related works section.

**Paper Type:**

methodological development

**Questions To Address In The Rebuttal:**

1. Explain why the confidence scores obtained from the method are not normalised to 0-1 range, whereas the scores for MC dropout and Deep Ensembles are normalised.
2. Show the trend of accuracy/number of slides eliminated with respect to the threshold
3. Justify lower variance of other methods in Fig 2.
4. Compare the uncertainty across different grades with a quantitative measure.

---

### Official Review · Reviewer_6VFE · 2023-02-04

**Confidence:** 4
**Preliminary Rating:** 4

**Summary:**

--The authors propose a new score to measure the confidence of AI models in grading tasks. The proposed method is simple and straight forward to understand. In addition, compared to other popular method, the proposed method is cost-effective.

--The proposed method is evaluated on their in-house dataset with the application on head and neck.

--The validation is sufficient.

**Strengths:**

--paper is well written and organized

--The proposed method is easy to examine the exist AI model without any extra training or inferences

--sufficient validation on a large dataset in three aspects, e.g. section 4.1, 4.2 and 4.3


**Weaknesses:**

major:

--no major weakness

minor:

--unclear for the application description (qualitatively), please consider to move Figure 3 to the main part. for section 3.1.1

--it is not clear for high/low confidence level in Table.1

**Deanonymize Review:**

no

**Detailed Comments:**

--please see weaknesses

--method is simple, lack of novelty. However, I think it could be a minor weakness or could not.

--shouldn't cite the in-house dataset for a double-blind review process.

**Paper Type:**

both

**Questions To Address In The Rebuttal:**

--why define the threshold as the median? please refer to the section 4.2

--what are the performance for other classes? please refer to section 4.1 or Figure 1.

--please explain why the proposed grade sensitive method is weakly supervised

---

### Meta-Review · Area_Chair_qWU5 · 2023-02-24

**Recommendation:** Accept (Poster)
**Confidence:** 5

**Metareview:**

The manuscript received two weak accept and one borderline recommendation; reviewers are in agreement that the method is easy to understand, clearly presented, and evaluated reasonably well.
Initial criticism that pertained to clarifications was well addressed, although the changes to the manuscript overall are quite minor.

From my perspective, it appears that the concern around using the median was not well addressed, as it leaves open the important demonstration on how to use this score in practice. While there is some reasoning about this in the response to reviewers, the manuscript does not adequately demonstrate the use of this method.
Similarly, the response to the lack of cross-entropy training is defensive; while it may be true that risk regression is more adequate here, demonstrating benefits over a standard baseline would have been a strength. This also seems to affect some claims around the generality of the method, as the baselines of ensembling and MC dropout are applicable to both classification and regression.

Overall, this is a borderline paper with moderate enthusiasm among reviewers.